# The Influence of Additives and Environment on Biodegradation of PHBV Biocomposites

**DOI:** 10.3390/polym14040838

**Published:** 2022-02-21

**Authors:** Pavel Brdlík, Martin Borůvka, Luboš Běhálek, Petr Lenfeld

**Affiliations:** Faculty of Mechanical Engineering, Technical University of Liberec, Studentska 1402/2, 46117 Liberec, Czech Republic; martin.boruvka@tul.cz (M.B.); lubos.behalek@tul.cz (L.B.); petr.lenfeld@tul.cz (P.L.)

**Keywords:** biodegradation, polyhydroxybutyrate-co-hydroxyvalerate, additives, thermophilic composting, freshwater biotope, vermicomposting

## Abstract

The biodegradation of polyhydroxybutyrate-co-hydroxyvalerate (PHBV) ternary biocomposites containing nature-based plasticizer acetyl tributyl citrate (ATBC), heterogeneous nucleation agents—calcium carbonate (CaCO_3_) and spray-dried lignin-coated cellulose nanocrystals (L-CNC)—in vermicomposting, freshwater biotope, and thermophilic composting have been studied. The degree of disintegration, differential scanning calorimetry (DSC), thermogravimetric analysis (TGA), and the evaluation of surface images taken by scanning electron microscopy (SEM) were conducted for the determination influence of different environments and additives on the biodegradation of PHBV. Furthermore, the method adapted from ISO 14855-1 standard was used for thermophilic composting. It is a method based on the measurement of the amount of carbon dioxide evolved during microbial degradation. The highest biodegradation rate was observed in the thermophilic condition of composting. The biodegradation level of all PHBV-based samples was, after 90 days, higher than 90%. Different mechanisms of degradation and consequently different degradation rate were evaluated in vermicomposting and freshwater biotope. The surface enzymatic degradation, observed during the vermicomposting process, showed slightly higher biodegradation potential than the hydrolytic attack of freshwater biotope. The application of ATBC plasticizers in the PHBV matrix caused an increase in biodegradation rate in all environments. However, the highest biodegradation rate was achieved for ternary PHBV biocomposites containing 10 wt. % of ATBC and 10 wt. % of CaCO_3_. A considerable increase in the degree of disintegration was evaluated, even in freshwater biotope. Furthermore, the slight inhibition effect of L-CNC on the biodegradation process of ternary PHBV/ATBC/L-CNC could be stated.

## 1. Introduction

One of the major problems of our time is the increasing pollution of the planet [1,2,3] and the reduction of non-renewable sources [4]. The consumption of plastic undoubtedly has a significant impact on this. The world demand for plastics has increased nearly twice as much in the last two decades (reaching 368 million tons in 2019) [5]. These environmental and social issues are linked to the growing population and the long half-life of standard petroleum-based plastics [6]. Consequently, the increasing interest of scientific and industrial researchers is focused on the developing materials with greater environmental sustainability, and biopolymers could be the solution. Although the production and usage of these materials are rapidly increasing, biopolymers currently still represent only about 1% of global production [7]. The biggest obstacle to the broader application of these materials is especially high production costs [8]. However, the use of appropriate feedstocks and innovation in processing technology offers the potential for further improvement [9]. In the content of the lifetime of biopolymers, it is important to know that not all bio-based polymers are biodegradable. In fact, 58.1% of biopolymer products are non-biodegradable [10]. The most commonly used biodegradable polymers include poly(lactic acid) (PLA), polyhydroxyalkanoates (PHA), and starch blends [10]. Since the packaging industry accounts for the greatest demand of plastic consumption (39.6%) [5], the largest potential use of these materials is also directed to this area. For the packing industry, health safety, barrier properties, mechanical and thermal properties are the most important parameters. From the published research [11,12,13,14], important health safety restrictions have been reported for PLA and PHA polymers. PHA are bacteria-synthesized biopolymers with a variety of physical and mechanical properties depending on types of monomers (150 different types), on the length of the side aliphatic chain at chiral carbon, on copolymerization, etc. [15]. From the group of PHA biopolymers for the packing sector has been reported poly(3-hydroxybutyrate-co-3hydroxvalverate) (PHBV) [13,14] as a suitable alternative. PHBV is a water-insoluble biopolymer with high surface energy [13]. This is compared to other biodegradable polymers with oxygen, UV and water vapor barrier properties [10,12]. It is currently used for food packaging with high oil content (marinated olives, cheese, nuts, etc.), and for frozen foods [13,14].

Regarding the summary above, biodegradability is a crucial parameter for reducing the environmental impact of the long half-life of single used packing products. According to the American Society of Testing and Materials (ASTM), the biodegradability of material is defined as “capable undergoing decomposition process into carbon dioxide, methane, water, inorganic compounds, or biomass in which the predominant mechanisms is the enzymatic action of microorganisms” [16]. Polymer materials are enzymatically degradable in a two-step. The first one consists of a reduction in polymer chains into low molecular weight oligomers, dimers, and monomers that are short enough to be assimilated by microorganisms in the second step [10]. Furthermore, the two mechanisms, surface and bulk erosion could be observed during degradation. The enzymes are relatively large particles (in comparison to chemicals and free radicals) unable to permeate the structure of polymers. Consequently, in the first degradation step, the enzymatic degradation occurs mainly at the surface [17,18]. In polyesters, such as polycaprolactone (PCL), poly(glycolic acid) (PGA), and PLA are the diffusion of water through the material (reaction with hydrolysable bonds) faster than hydrolytic attack [17]. Hence, bulk degradation is a typical mechanism for the defragmentation of these materials. On the contrary, the diffusion of water (a bulk mechanism) is, in PHBV, relatively slow, and therefore, the dominant degradation mechanism is surface erosion [17,19,20]. The biodegradation rate depends on many factors. One of the most important factors is the chemical nature of the chain, its molecular weight, and distribution [10,17]. Tokiwa et al. [21] reported faster enzymatic degradation reaction in PHBV copolymers than in homopolymers (P3HB, P3HV). Furthermore, Weng et al. [22] found higher degradation in PHBV biopolymers containing higher mol content of HV. The presence of HV units accelerates the hydrolysis process and decreases the crystallinity of the materials [23]. The high crystallinity is sometimes ascribed as a reason behind the slow biodegradation of PHAs [24]. This is because water molecules easily diffuse into amorphous regions, and are easily assimilated by microorganisms. Sevim et al. [25] and Herzog et al. [26] reported the effect of molecular weight on the biodegradation process of PHBV. Faster hydrolytic and enzymatic degradation rates were observed in polyester with a lower molecular weight. In a study, Buchholz et al. [27] increased resistance to enzymatic degradation for polyester with side chains and lover molecular weight. Hence, aside from molecular weight, the side chains and stereoregular conformation that affect chain flexibility and mobility have to be considered. As was mentioned, hydrolytic and enzymatic degradation are typical mechanisms for the biodegradation process of polyesters. Consequently, the environment and medium characteristics such as temperature, pH, and the presence and flow of oxygen are tremendously important factors. Therefore, plenty of works have been dedicated to this issue. Luo et al. [19] reported significant changes of the mechanical properties and weight reduction of PHBV exposed to composting in 50 days. Mergaert et al. [28] evaluated different weight reductions of PHBV copolymers (10 and 20% of HV) in soils at 28 °C, which were characteristic with different pHs. The basic groups and chemicals in soils caused the faster hydrolysis of ester groups; faster than acidic ones. Hence, faster degradation was observed. Weng et al. [22,29] reported a high level of biodegradation in the thermophilic condition of pilot-scale composting and controlled composting. When the polyesters are exposed to higher temperature than the glass transition temperature, the flexibility of the chain is increased. Consequently, the polyesters are easily facilitating the hydrolysis reaction and the attachment of microbes/enzymes [17,30]. Furthermore, Muniyasamy et al. [31] evaluated the faster biodegradation of PHBV under composting conditions than under burial conditions. Compared to composting, the slightly lower biodegradation of PHBV samples exposed to the sea and river water (marine) was evaluated in several studies [20,32,33,34]. On the contrary, very intensive degradation was observed in an anaerobic sludge environment [18,35,36].

The biodegradation of neat PHBV is a relatively well-known issue. There are also some studies focused on the evaluation of the influence of natural fillers, such as pomace filler [37], wine shoots [38], miscanthus and dried grains [39], on the biodegradation of PHBV. However, the influence of additives that can improve chain flexibility, mobility, or increase the crystallization kinetics of PHBV, has still not been sufficiently explained. Several modifications can be used to enhance chain flexibility and mobility [40,41]. In the packaging industry, there is often the addition of plasticizers [42,43,44]. Plasticizers are an effective way to reduce brittleness. However, the enhancement of chain flexibility and mobility is very often accomplished with a decrease in mechanical, thermal, or barrier properties [45]. On the contrary, the application of nucleation agents could, due to an increasing crystallinity degree, evoke increasing mechanical, barrier properties, and thermal resistance. However, the increased degree of crystallinity also evokes an increase in brittleness. The synergistic effect of heterogeneous nucleation and increased chain mobility could be the solution [46]. Therefore, the current work is dedicated to the evaluation of the influence of nature-based plasticizer acetyl tributyl citrate (ATBC), heterogeneous nucleation agents based on calcium carbonate (CaCO_3_), and spray-dried lignin-coated cellulose nanocrystals (L-CNC) on the biodegradation behavior of PHBV. Regarding the previous summary, the biodegradation process is considerably influenced by medium characteristics. Hence, three different biodegradation tests were performed. The first one deals with the determination of the biodegradation potential of PHB bio-composites and ternary composites in the thermophilic composting conditions. The second one was focused on the analysis of biodegradability during the process of vermicomposting. It is an environmental characteristic with high moisture content and mesophilic temperatures. The last degradation experiment was for the determination effect of hydrolytic attacks made in the freshwater biotope.

## 2. Materials and Methods

P3-HB 3-HV (3% mol HV) (PBHV), in the form of pellets (Enmat Y1000P), was provided by TianAn Biopolymer (Ningbo, China). It is a biopolymer made from the cupriavidus necator fermentation of D-glucose and propionic acid, with melt temperature in the range of 160 °C to 175 °C, glass transition temperature of 8 °C, density of 1.24 g∙cm^−3^, and the average molecular mass of 485,000 g∙mol^−1^. Furthermore, natural citric acid raw material (ATBC) derived from the fermentation of corn under the trade name of Citroflex A-4 (Vertellus Holding LLC, Indianapolis, IN, USA) was used in the form of an oily liquid as a plasticizer. Spray-dried L-CNC (BioPlus-LTM Crystals) purchased from American Process Inc. (Atlanta, GA, USA) and precipitated calcium carbonate (CaCO_3_) (Honeywell Fluka, Seelze, Germany) were used as a nucleation agent. L-CNC is characteristic, with an average particle size of 4–5 nm in width, and 500 nm in length. The average particle size is lower than 1 µm in precipitated calcium carbonate (CaCO_3_).

### 2.1. Preparation of PHBV Films

The PHBV pellets, as well as CaCO_3_ and L-CNC additives, were dried for 24 h at 50 °C in vacuum oven VD53 (Binder GmbH, Tuttlingen, Germany), before being processed to remove eventual moisture. The laboratory micro-compounder MC 15 HT (Xplore, Netherlands) equipped with conical screws, at a speed of 100 rpm, and a constant temperature profile of 180 °C, was used as the primary processing device to compounding plasticized PHBV and ternary composite granulates. The individual compositions of material variants are listed in Table 1. Compounded materials were again dried (24 h, 50 °C) and further extruded on the twin-screw extruder MC 15 HT trough-connected flat film die, with a 0.4 mm gap size. All films were extruded at melt temperature of 190 °C, 80 rpm screw speed, and drawn with cooled rolls under a 1.1 speed ratio.

### 2.2. Analysis of Biodegradability under Thermophilic Composting

The biodegradability analysis under thermophilic composting conditions was carried out according to a method adapted from ISO 14855-1 standard. This method is based on the measurement of the amount of carbon dioxide evolved during microbial degradation. For this purpose, spirometer ECHO (ECHO d.o.o., Slovenske Konjce, Slovenia) with automatic leak detection and automatic humidification was used. The compost purchased from AGRO CS (Říkov, Czech Republic) is characteristic, with 6.5 pH (measured by the Voltcraft PH-100ATC pH meter) (VOLTCRAFT, Wollerau, Switzerland), and 26.6 g content of volatile solids (evaluated by 550°/5 h in the oven CLASIC 3014) (CLASIC CZ, Řevnice, Czech Republic) was used in this study. Regarding the standard, pebbles and any foreign objects larger than 2 mm were removed from the compost, and the 50% humidity water content was adjusted in compost by the halogen moisture analyzer Mettler Toledo™ HX204 (Mettler Toledo, Columbus, OH, USA). Furthermore, the tested PHBV films (10 g) were trimmed to individual pieces with sizes of about 1 cm × 1 cm, placed to 2.8 L cylindrical vessels, together with 150 g of compost and hermetic closed. Each biodegradation analysis was performed in duplicity at a constant temperature of 58 °C. The control blank vessels and vessels with microcrystalline cellulose (Sigma-Aldrich, Saint-Quentin-Falavier, France) were used as the reference and control materials of the proper microbiological activity of the compost. The respirometers were shielded from the light. Every week (for a 3-month period), glass vessels were opened, and the inoculum was stirred to ensure an even distribution of moisture.

During biodegradation measurement, it is not possible to take any samples (change content of organic carbon). Consequently, any further analysis could not be evaluated from this experiment. Therefore, a parallel investigation was carried out, where the PHBV films with the size of 100 mm × 35 mm were exposed to the same condition (58 °C, 50% humidity water content), as in the experiment presented above. The PHBV films were controlled every week. Upon the significant disintegration of PHBV films, the experiment was terminated after 30 days. Furthermore, thermal analysis (DSC), thermal degradation (TGA), surface changes, and degree of disintegration were, after conditioning (25 °C/96 h), evaluated. The results from the introduced analyses were evaluated from five measurements. Due to the small data set, only average values were provided, and thus standard deviation was not specified.

### 2.3. Analysis of Biodegradability under Vermicomposting (Mesophilic Condition)

The process of the vermicomposting of household waste (coffee grounds, potato peelings, carrots, and other vegetables) by the Epigeic earthworm Eisenia foetida was applied to determine its influence on PHBV bio-composites and ternary composites biodegradation rate. The vermicompost environment was characterized by pH 8.2 (measured by Voltcraft PH-100ATC pH meter), the ambient temperature of 21 °C, and 81% moisture content (Mettler Toledo™ HX204). The films with a size of 100 mm × 35 mm were exposed to the biodegradation process for 30 days. Subsequently, films (25 °C/96 h), evaluated thermal properties (DSC), thermal degradation (TGA), surface changes, and calculated degree of disintegration were evaluated. The results were, as in thermophilic composting, evaluated from five measurements. Due to the small data set, only average values were provided, and thus standard deviation was not specified.

### 2.4. Analysis of Biodegradability in Freshwater Biotype

The biodegradation of plasticized PBHV and ternary bio-composites was also studied in the freshwater biotope. The films with a size of 100 mm × 35 mm were placed in a steady biotope (aquatic plants, algae, fish, snails, etc.), which was characteristic, with pH 7.8 and temperature 27 °C. As in an analysis of biodegradability under vermicompost environment, there were films after 30 days exposition, drawn and conditioned (25 °C/96 h). The degree of disintegration, thermal properties (DSC), thermal degradation (TGA), and surface changes was evaluated (5 measurements) from conditioned films, and only average values were provided.

### 2.5. Calculation of the Percentage Biodegradation

The percentage of biodegradation was determined from the cumulative amount of released carbon dioxide (CO_2_), in accordance with the following equation:(1)$$D_{t}=\frac{{\left({\mathrm{CO}}_{2}\right)}_{T}-{\left({\mathrm{CO}}_{2}\right)}_{B}}{T_{{\mathrm{hCO}}_{2}}}\cdot 100$$
where (CO_2_)_T_ is the cumulative amount of carbon dioxide evolved in the composting vessel containing the test material, (CO_2_)_B_ is the mean cumulative amount of carbon dioxide evolved in the blank vessels, and (T_hCO2_) is the theoretical amount of carbon dioxide that can be produced by the test material (all in g/vessel).

The theoretical amount of carbon dioxide can be determined via the following equation:(2)$$T_{{\mathrm{hCO}}_{2}}=M_{\mathrm{TOT}}{*C}_{\mathrm{TOT}}\cdot \frac{44}{12}$$
where M_TOT_ is the total number of dry solids in the test material introduced into the composting vessel at the start of the test (in g), C_TOT_ is the proportion of total organic carbon in the total dry solids in the test material (in g/g), and 44 and 12 are the molecular mass of carbon dioxide and the atomic mass of carbon, respectively. The individual proportions of total organic carbon of PHBV biocomposite components are listed in Table 2.

### 2.6. Calculation of Degree of Disintegration

The degree of disintegration was calculated in accordance with the following equation:(3)$$D=\frac{m_{i}-m_{r}}{m_{i}}\cdot 100$$
where D, is degree of disintegration (in %), m_i_ is initial dry mass of the test PHBV films (in g), m_r_ is the dry mass of the residual test material (in g).

### 2.7. Differential Scanning Calorimetry (DSC)

A Mettler Toledo DSC 1/700 calorimeter (Mettler Toledo, Greifensee, Switzerland) was used to measure cold crystallization temperatures and enthalpies (T_cc_, ∆H_cc_), the determination of melting temperatures and enthalpies (T_m_, ∆H_m_), and the evaluation changes of degree of crystallinity X_c_. The degree of crystallinity was calculated through the following equation, where ∆H^0^_m_ is the melting enthalpy of 100% crystalline PBHV (146.6 J∙g^−1^) [47], and w_m_ is the mass fraction of PHBV in the composites. Samples test procedures were as follows. Samples of approximately 5 mg were taken from the same location of PHBV films, placed in an aluminum pan, and sealed. The used program was: heating from −40 °C to 200 °C with a heating rate of 10 °C∙min^−1^, isotheral hold for 180 s to remove previous thermal history, and then cooled again from 200 °C to 25 °C (10 °C∙min^−1^).
(4)$$X_{C}=\frac{\Delta H_{m}}{\Delta H_{m}^{O}\cdot w_{m}}\cdot 100$$

### 2.8. Thermogravimetric Analysis (TGA)

Thermal degradation was performed with a thermogravimetric analyzer TGA2, Mettler Toledo (Mettler Toledo, Greifensee, Switzerland). The taken samples (5 ± 0.5 mg) were heated from 50 °C to 600 °C under an N_2_ atmosphere at the heating ramp of 10 °C∙min^−1^. Thermal characteristics, such as decomposition temperature determined at 3% weight loss (T_3%_) and 50% weight loss (T_50%_), were further evaluated.

### 2.9. Scanning Electron Microscopy (SEM) Images

The surface changes of PHBV films were observed using field emission scanning electron microscope (FE-SEM) Carl Zeiss ULTRA (Carl Zeiss, Oberkochen, Germany) under an accelerated voltage of 3 kV. The test samples were coated with 1 nm of platinum using Q150R ES (Quorum Technologies, Lewes, UK).

## 3. Results and Discussion

### 3.1. Determination of Ultimate Aerobic Biodegradability

The courses of biodegradation curves are shown in Figure 1. The production of carbon dioxide in a blank control vessel showed 120 mg of CO_2_ per gram of volatile solids during the first 10 days, and the biodegradation degree of reference microcrystalline cellulose (MCC) powder sample reached 82% after 45 days. Based on these results, the proper microbial activity of the compost could be claimed, and the validity of the results (ISO 14855-1) could be confirmed. Significant differences in biodegradation courses could be observed. The highest biodegradation rate showed PHBV ternary composite films containing 10 wt. % of ATBC and CaCO_3_. The induction period of the ternary composite was 6 days shorter when compared to other PHBV films. Very similar biodegradation dependence was observed in the first 30 days for neat PHBV, PHBV/ATBC, and PHBV/ATBC/L-CNC films. The level of biodegradation was 32% for neat PHBV, 35% for plasticized PHBV; 33% for ternary bio-composite PBHV/ATBC/L-CNC. On the contrary, the degree of biodegradation for PHBV/ATBC/CaCO_3_ reached 56%. Since then, the plasticized PHBV/ATBC showed the higher degradation rate. The starting plateau phase was, for PHBV/ATBC films, achieved 14 days earlier than for PHBV/ATBC/L-CNC films, and 42 days earlier than for neat PHBV. However, the level of biodegradation was for neat PHBV, PHBV/ATBC, and PHBV/ATBC/L-CNC, similar after 90 days. Thus, the difference between the tested variations is not in the level of achieved microbial degradation, but in its rate. Based on the results, all material variants that accomplished the requirements of EN 13432 standard for packaging recoverable through composting can be stated.

In our previous study [48], we reported the similar dependence of the influence of ATBC, L-CNC, and CaCO_3_ additives on the biodegradation potential of PLA. The synergistic effect of ATBC/CaCO_3_ on the biodegradation rate PLA has been observed. Husarova et al. [49] reported that the presence of calcium carbonate additives has a positive influence on the bioavailability of the biodegradable compounds of the material in compost and soil environments. Furthermore, Mohammadi et al. [50] reported the enhancement of the soil burial biodegradation of the biopolymer by employing CaCO_3_ particles. The biodegradability of PLA/CaCO_3_ biocomposites was enhanced with the increasing content of CaCO_3_. The reason behind this phenomenon has been ascribed to the release of CaCO_3_ in the biodegradation medium, which offers buffering action, and helps to prevent soil acidification [51,52]. Acidic environments are unsuitable for many microbes and enzymes’ activities [18]. Suharty et al. [53] ascribed the enhancement biodegradability of PP matrix with nano-sized CaCO_3_ to increasing water absorb ability. Another reason could be a surface treatment of CaCO_3_, with fatty acids that is used very often (1–3 wt. %) for increasing the dispersibility of particles in the polymer [54]. The fatty acid causes a chemical reaction with the ester linkage, which evokes chain scission, and induces thermal degradation [55]. However, signs of fatty acids have not been observed during the TGA and FT-IR analysis (see Appendix A). Furthermore, the inhibition effect of the CNC lignin coating on the biodegradation of PLA ternary bio-composites was observed. Yang et al. [56] and Micales et al. [57] reported that lignin inhibits the degradation of cellulose. However, there are obvious differences between the experiments in the rate of biodegradation. PHBV bio-composites showed faster biodegradation rates. The plateau phase (which represents the end of biodegradation) is for the PHBV/ATBC/CaCO_3_ ternary composite achieved in 57 days, and for PHBV/ATBC in 65 days. On the contrary, the biodegradation for the same combination of additives in PLA bio-composites was still unfinished after 75 days. When the biodegradation rates of neat PLA and PHBV are compared, 42% and 87% of the biodegradation level were achieved after 75 days. This confirms the high biodegradation potential of PHBV exposed to thermophilic composting that was reported by Salomez et al. [10] and Weng et al. [29]. Therefore, the higher influence of used additives on biodegradation rate was under thermophilic composting conditions observed for PLA than for PHBV-based matrix biocomposites. A comparison of the biodegradation levels of both experiments within 75 days is shown in Table 3.

### 3.2. Degree of Disintegration

The degree of disintegration results for PHBV biocomposites and ternary composites exposed to vermicomposting, freshwater biotope and the thermophilic composting for 30 days is shown in Figure 2. The highest level of disintegration was achieved by the thermophilic composting conditions, where full disintegration was detected after 45 days. The same observation has been reported by Weng et al. [22,29]. The full disintegration of neat PHBV films in thermophilic conditions of pilot-scale composting and laboratory-scale composting within 35 days has been observed. Boonmee et al. [36] observed a slower disintegration rate of neat PHBV films exposed to thermophilic oxygen-limited conditions. The full disintegration was achieved after burial exposure within 75 days. Volova et al. [58] reported a 43.5% weight reduction of PHB within 42 days’ exposition to a small recreational eutrophic reservoir. The evaluated level of disintegration of PHBV films exposed to freshwater biotope confirmed the lower biodegradation potential of this environment. The neat PHBV films showed 10% mass loss. The PHBV/ATBC and PHBV/ATBC/L-CNC bio-composites showed 15% and 14% weight reduction, respectively. Slightly higher disintegration was observed in the vermicompost environment. The difference is particularly noticeable, especially in a PHBV ternary composite with ATBC and CaCO_3_. During vermicomposting, the 39% mass loss was evaluated, contrary to 22% mass loss in the freshwater biotope. These results confirm the results of Sadami et al. [17], where surface enzymatic degradation is declared as the dominant degradation mechanism of PHBV. In accordance with the results of biodegradation (ISO 14855-1), the lowest disintegration was evaluated in neat PHBV. The addition 10 wt. % of ATBC plasticizer ensured the increase in biodegradation potential. Thsou et al. [59] reported a rising water absorption rate with increasing content of ATBC in PLA, which could evoke faster hydrolysis and higher microbial enzyme activity. The inhibition effect of the lignin coating of CNC on the biodegradation of PHBV ternary bio-composites could not be confirmed, due to low differences of degree of disintegration. The most significant effect on disintegration level has been observed after the addition of CaCO_3_, where the disintegration was at a relatively high level, even in environments with lower biodegradable potential.

### 3.3. Differential Scanning Calorimetry (DSC)

The results of the first non-isothermal heating of PHBV films are reported in Table 4, and shown in Figure 3. Maiza et al. [60] and Courgneau et al. [61] reported a significant decrease in glass transition (T_g_) and melt transition (T_m_) temperatures of PLA polyester due to ATBC addition. On the other hand, Martino et al. [62] observed a lower shift of transition temperatures in PHBV bio-composites containing ATBC plasticizers. The reason for the different plasticizer efficiency of both polyesters could be found in the morphologic structure. This plasticizes the increase mobility of polymer chains (increase intermolecular distance) in the amorphous phase. Consequently, lower changes of thermal and mechanical properties could be expected more for high-crystalline PHBV than for PLA, which is characteristic, with a slow crystallization rate and a less ordered structure. Our results confirmed this statement. Only a slight decrease in melt (T_m_) and cold crystallization temperatures (T_cc_) was observed in plasticized PHBV/ATBC. Furthermore, the glass transition temperature (T_g_) is not due to the high crystalline content visible. Hence, it could not be reported. However, the increase in chain mobility evoked the enhancement of crystallinity degree (X_c_). Aframehr et al. [50], Kirboga et al. [63,64], and Wang et al. [65] reported the nucleation effect of both micro- and nano-CaCO_3_ particles to the crystallization behavior of PLA, and its influence on the enhancement of themo-mechanical properties. Gupta et al. [66] evaluated the nucleation efficiency of L-CNC on the PLA crystallization. The results of our previous study [67] showed a synergic effect of 1 wt. % L-CNC and 10 wt. % ATBC on the crystallization of the PLLA matrix during injection molding. Therefore, further crystallinity degree (X_c_) enhancement was expected in PHBV/ATBC/L-CNC ternary bio-composite systems. Nevertheless, no significant changes of crystallinity degree or transient temperatures were noticed for PHBV/ATBC/L-CNC ternary bio-composite, when compared to neat PHBV. The results of PHBV/ATBC/CaCO_3_ ternary composites showed a similar trend. The fast cooling of thin, extruded films could be ascribed to this phenomenon. In the process of injection molding, the more favorable conditions for crystallization could evoke the greater influence of nucleation agents and increased chain mobility on the degree of crystallinity (X_c_). Furthermore, bimodal endothermic peaks have been observed at DSC curves during the first heating cycle. This observation could be caused by being ascribed to different morphology (lamellar thickness, distribution, perfection, or stability) and the physical aging of the rigid amorphous phase fraction [68,69]. PHBV films during the second heating cycle showed single endothermic peaks (data not shown). Consequently, the influence of process condition (intensive cooling) on the structure formation of PHBV films can be confirmed.

When the results of DSC analyses (the first heating cycle) of the neat PHBV films exposed to 30 days biodegradation are compared, the films exposed to vermicomposting and thermophilic composting conditions showed a slight decrease in the degree of crystallinity (X_c_). The differences in transition temperatures are due to the high crystallinity degree of PHBV, which is very small. The changes of bimodal to single endothermic peaks were further observed in PHBV bio-composites and ternary composites, containing ATBC plasticizers after vermicomposting. Consequently, the structure changes and development of the more stable and uniform crystalline structure could be assumed by higher PHBV chain mobility and the environment condition of vermicomposting. These structure changes could lead to changes in the degree of crystallinity. This crystallinity enhancement is apparent, especially in PHBV/ATBC/L-CNC and PHBV/ATBC/CaCO_3_ ternary bio-composites. However, the structure changes did not evoke any significant changes in melt temperature and cold crystallization temperatures. The distribution of bimodal peaks of PHBV films exposed to freshwater biotope and thermophilic composting showed the enhancement of structural differences. It means that hydrolytic and enzymatic degradation did not affect all morphological parts equally, and the biodegradation process in these environments is influenced by shape, the thickness of lamellas, and their distribution, perfection, or stability. The greatest differences in the shape of bimodal peaks compared to initial state showed PHBV films exposed to thermophilic composting, thus the environment with the highest reported biodegradation and disintegration rates. In the transient temperatures and degree of crystallinity of PHBV, bio-composites exposed to freshwater biotope and thermophilic composting environment did not show any significant changes. However, it is important to note that the crystallinity degree (X_c_) calculated for PHBV films exposed to thermophilic composting has been due to intensive disintegration, with no guarantee of the exact weight content of additives, and only has an informative character.

### 3.4. Thermogravimetric Analysis (TGA)

The results of TGA are shown in Figure 4 and summarized in Table 5. The significant influence of additives on the decomposition temperatures of PHBV films has been observed. The presence of 10 wt. % of ATBC plasticizer caused a decrease in thermal stability. Initial degradation temperature (T_3_%) dropped when compared to neat PHBV by ~17 °C, and temperature at the midpoint (T_50_%) by ~ 4 °C. Erceg et al. [70] evaluated a similar decrease in decomposition temperatures for PHB/ATBC biocomposites that were made by compression molding. Moreover, Maiza et al. [59] reported a significant decrease in thermal stability for PLA polyester-bio-composites containing ATBC plasticizer. However, the decrease in decomposition temperatures was higher for PLA bio-composites than in our PHBV bio-composites. Zhang et al. [71] reported a considerable decrease in initial decomposition temperature for the ATBC-plasticized polyurethanes. As was stated in the DSC section, the low concentration (1 wt. %) of L-CNC did not increase the degree of the crystallinity of PHBV films. Hence, any increase in thermal stability of PHBV/ATBC/L-CNC bio-composites has been observed. The highest decrease in thermal stability (initial decomposition temperature) has been observed in PHBV/ATBC/CaCO_3_ ternary bio-composite systems. Initial degradation temperature (T_3_%) decreased by around 33 °C, and temperature at the midpoint (T_50_%) by around 22 °C. A similar trend has been reported by Nekhamanurak et al. [55], where a negative impact of the fatty acid treatment of CaCO_3_ on the thermal stability of polyester-based bio-composites has been observed. However, the presence of fatty acid was not evident in used, precipitated CaCO_3._ On the contrary, Kirboga et al. [63,64] reported the increased thermal stability of PHBV/CaCO_3_ biocomposites, which was evoked by the nucleation effect of CaCO_3_. Regarding the previous summary of thermal analyses, where any significant nucleation effect was not detected, the decrease in thermal stability could be ascribed to interaction between ATBC plasticizer and CaCO_3_.

When the decomposition temperatures (T_3_% and T_50_%) of PHBV bio-composites exposed to different environments are compared, the highest decrease was evaluated in PHBV films exposed to thermophilic composting. Consequently, the highest biodegradation rate of this environment can be stated. Neat PHBV, as well as PHBV/ATBC and PHBV/ATBC/L-CNC bio-composites exposed to the freshwater biotope, showed no, or only a minimal decrease, in decomposition temperatures (T_3_% and T_50_%). However, PHBV ternary composites containing CaCO_3_ showed a considerable decrease in decomposition temperatures in all environments (as well as weight reduction). It means that CaCO_3_ evokes the enhancement of the hydrolytic (bulk) mechanism, as well as the enzymatic mechanism of degradation. When the decomposition temperatures of PHBV films exposed to the vermicomposting and freshwater environment are compared, a slightly higher decrease in thermal stability was observed for PHBV films exposed to the vermicomposting. These results declare the higher tendency of PHBV to enzymatic surface degradation than to the hydrolytic bulk one [17]. From the evaluated degradation temperatures above, we state our conclusions on the influence of additives on the process of the biodegradation of PHBV. The addition of ATBC plasticizers causes a decrease in decomposition temperatures (compared to neat PHBV). A slightly lower decrease was observed for PHBV containing lignin-coated CNC. Furthermore, PHBV/ ATBC/ CaCO_3_ ternary bio-composites showed the highest decrease in decomposition temperatures.

### 3.5. Scanning Electron Microscopy (SEM) Images

The SEM images of PHBV film surfaces after processing and by the end of vermicomposting, thermophilic composting and freshwater biotope expositions, are shown in Figure 5, Figure 6 and Figure 7. All produced PHBV films showed smooth surfaces. However, apparent differences are evident on the surface images of PHBV films exposed to different environments after 30 days. The PHBV films after degradation in vermicomposting showed eroded surfaces without any pinholes. Similar results of enzymatic surface degradation of PHBV were reported by Weng et al. [29] and Boonmee et al. [36]. The increase in surface erosion in plasticized PHBV is further evident from surface images. It demonstrates a higher microbial enzymatic degradation rate. Consequently, it could be stated that increasing molecular mobility via the addition of ATBC plasticizer increases surface erosion, as well as microbial enzymatic degradation. The structural changes were reported in DSC section, where a course of bimodal to single endothermic peaks has been observed. When the surface images of PHBV films after degradation in vermicompost environment are compared with other degradation environments, the uniform spherulitic structure is clearly visible. Hence, the structure changes could be confirmed. Due to slightly lower roughness when compared to plasticized PHBV/ATBC, the inhibition effect of L-CNC could also be confirmed. The highest erosion, which subsequently led to increased roughness, has been observed in PHBV ternary composites containing CaCO_3_. The combination of ATBC plasticizer and CaCO_3_ evoked a further increase in surface erosion that ensured the disintegration of PHBV into fragments.

The PHBV films exposed to freshwater biotope showed a relatively smooth surface with many pinholes. Consequently, another degradation mechanism could be assumed. The enzymatic degradation that caused uniform surface degradation was observed in PHBV films after the process of vermicomposting. On the contrary, the freshwater biotope environments increased the hydrolytic attack (a bulk mechanism), which leads to pinholes evolution. The higher number of pinholes than for neat PHBV has been detected in the SEM imagines of plasticized PHBV/ATBC films. The number of pinholes decreased in PHBV/ATBC/L-CNC bio-composites. Hence, the above-mentioned influence of plasticizer on the enhancement of biodegradation rate, as well as the inhibition effect of L-CNC, can be confirmed for freshwater biotope environments. The largest surface changes (increased surface roughness and size of pinholes) were observed in the PHBV/ATBC/CaCO_3_ material combination.

The SEM surface imagines of PHBV films exposed to thermophilic composting for 30 days showed the disintegration of PHBV into fragments. The releasing of fragments from the surface of the neat PHBV films is clearly seen in Figure 7b. When the defragmentation (decreasing of weight and thickness) achieves critical sample thickness, the formation of cracks occurs. The propagation of cracks further evoked pinholes creation. Hence, the rough surface and pinholes could be observed on the surface of PHBV films. The highest defragmentation, the highest roughness, and the largest pinholes were observed in PHBV ternary bio-composite containing ATBC plasticizer and CaCO_3_. Consequently, the highest degradation rate could be claimed for this material combination. On the contrary, the lowest level of defragmentation was observed in neat PHBV. The inhibition effect of lignin coating of CNC is due to the intensively degradation process, not obvious when comparing PHBV films exposed to vermicomposting and freshwater biotope.

## 4. Conclusions

The influence of biobased plasticizer acetyl tributyl citrate (ATBC), heterogeneous nucleation agents based on calcium carbonate (CaCO_3_) and spray-dried lignin-coated cellulose nanocrystals (L-CNC) on the biodegradation behavior of PHBV in vermicomposting, freshwater biotope, and thermophilic composting has been studied. The highest biodegradation rate was observed in a thermophilic composting environment. The biodegradation level (ISO 14855-1) of all PHBV-based samples was, after 90 days, higher than 90%. Due to this, the requirements of EN 13432 standard for packaging recoverable through composting and biodegradation could be accomplished. The thermophilic condition of composting ensured the intensive disintegration of PHBV films into fragments (full disintegration within 45 days), which evoked an apparent decrease in decomposition temperatures (T_3_% and T_50_%). The lowest disintegration level and the lowest changes in decomposition temperatures of PHBV films were observed in freshwater biotope. The vermicomposting conditions showed slightly higher biodegradation potential. Furthermore, the different mechanisms of degradation are apparent from evaluated SEM surface images. PHBV films exposed to vermicomposting for 30 days showed, due to surface enzymatic degradation, relatively uniform eroded surfaces, where the spherulitic structure was evident. The freshwater biotope environment increased hydrolytic attack to PHBV films, which evoked the creation of many pinholes. The high disintegration intensity during thermophilic degradation caused the surface (enzymatic) degradation to achieve critical sample thickness, and the formation of cracks and pinholes occurs within the same exposition time.

The application 10 wt. % of ATBC plasticizers into PHBV matrix caused a significant decrease in thermal stability, as well as a shift in transient temperatures and an increased degree of crystallinity. The enhancement of chain mobility further evoked an increase in the biodegradation rate of PHBV. The synergic influence of both additives of PHBV/ATBC/CaCO_3_ ternary bio-composite on the enhancement of the degree of crystallinity and the shifting of transient temperatures was not achieved. The fast cooling of thin extruded films could be ascribed to this phenomenon. However, the exponential increase in biodegradation potential was for PHBV/ATBC/CaCO_3_ observed. Considerable increase in the degree of disintegration was evaluated, even in freshwater biotope. Moreover, in PHBV ternary bio-composites, the synergic effect of ATBC (10 wt. %) and L-CNC (1 wt. %) on the crystallization was not observed due to fast cooling. Furthermore, the slight inhibition effect of L-CNC on the biodegradation process of ternary PHBV/ATBC/L-CNC could be stated. The similar influence of ATBC, L-CNC and CaCO_3_ additives to biodegradation potential of PLA was observed in our previous study. However, there are obvious differences between PLA and PHBV biocomposites and ternary composites in the rate of biodegradation. The PHBV films showed under thermophilic composting condition faster biodegradation rates. Consequently, a higher influence of used additives on biodegradation was observed for PLA than for PHBV biocomosites and ternary composites.

## Figures and Tables

**Figure 1 polymers-14-00838-f001:**
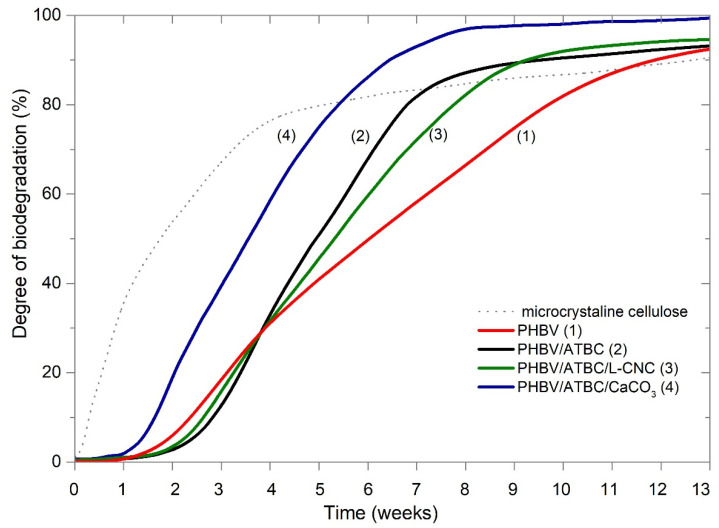
Biodegradation curves of PHBV biocomposites under controlled composting ISO 18455-1.

**Figure 2 polymers-14-00838-f002:**
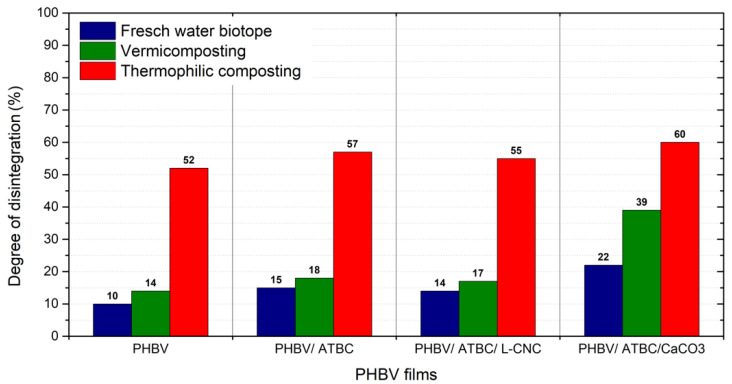
Degree of disintegration of PHBV bio-composites after 30 days’ degradation.

**Figure 3 polymers-14-00838-f003:**
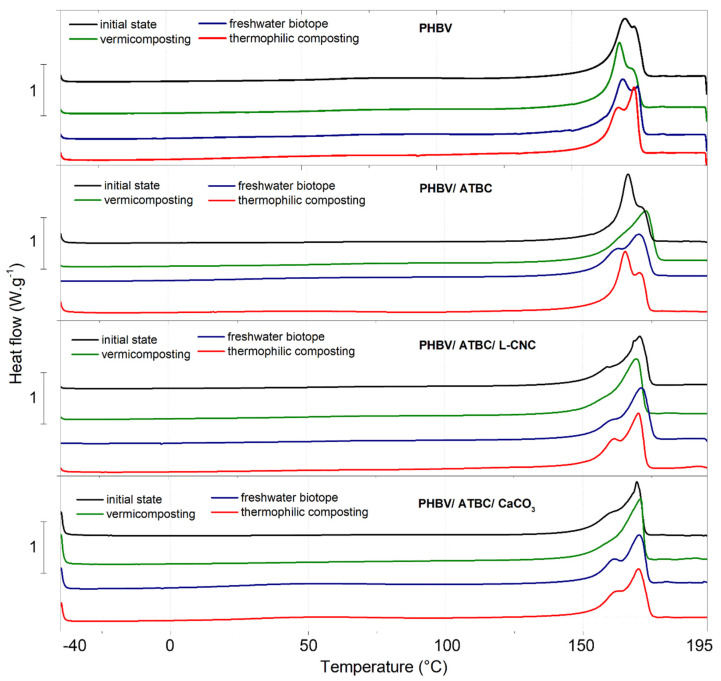
Thermal analysis (DSC) curves of PHBV bio-composites at initial state and after 30 days degradation.

**Figure 4 polymers-14-00838-f004:**
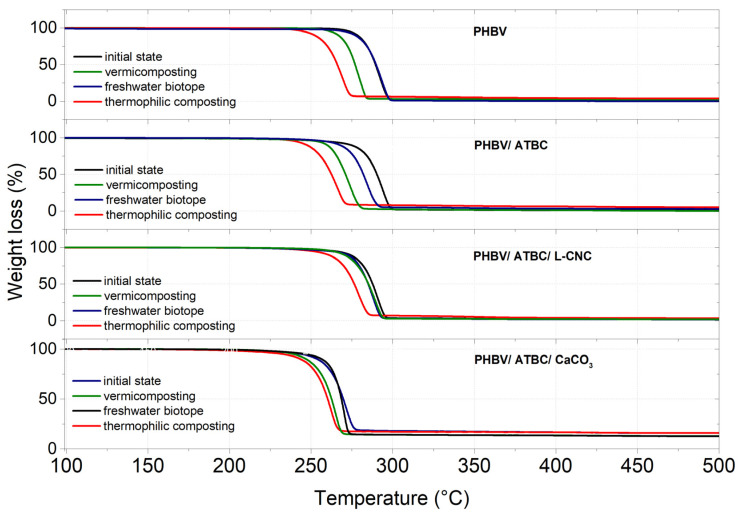
Thermogravimetric analysis (TGA) curves of PBHV bio-composites at initial state, and after 30 days of degradation.

**Figure 5 polymers-14-00838-f005:**
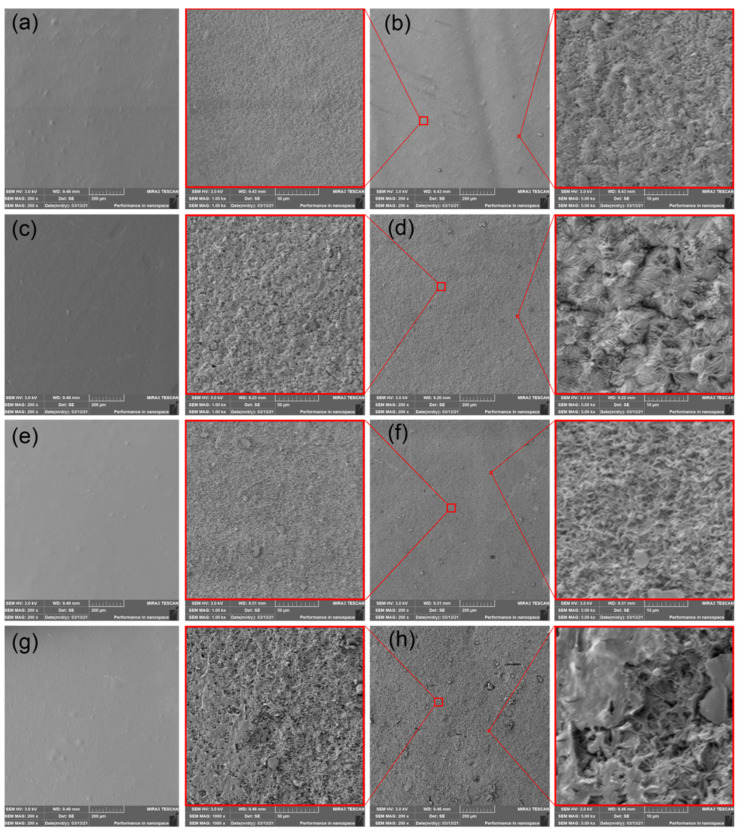
SEM images of (**a**) neat PHBV film at initial state, (**b**) neat PHBV film after 30 days of degradation in vermicompost environment, (**c**) PHBV/ATBC at initial state, (**d**) PHBV/ATBC film after 30 days of degradation in vermicompost environment, (**e**) PHBV/ATBC/L-CNC at initial state, (**f**) PHBV/ATBC/L-CNC film after 30 days of degradation in vermicompost environment, (**g**) PHBV/ATBC/CaCO_3_ at initial state, (**h**) PHBV/ATBC/CaCO_3_ film after 30 days of degradation in vermicompost environment.

**Figure 6 polymers-14-00838-f006:**
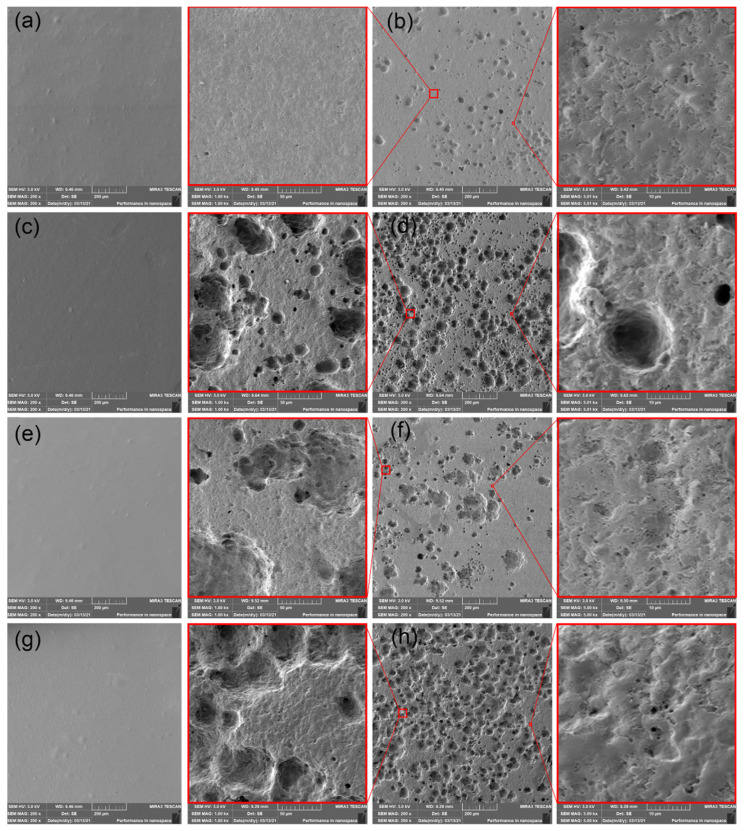
SEM images of (**a**) neat PHBV film at initial state, (**b**) neat PHBV film after 30 days of degradation in freshwater biotope, (**c**) PHBV/ATBC at initial state, (**d**) PHBV/ATBC film after 30 days of degradation in freshwater biotope, (**e**) PHBV/ATBC/L-CNC at initial state, (**f**) PHBV/ATBC/L-CNC film after 30 days of degradation in freshwater biotope, (**g**) PHBV/ATBC/CaCO_3_ at initial state, (**h**) PHBV/ATBC/CaCO_3_ film after 30 days of degradation in freshwater biotope.

**Figure 7 polymers-14-00838-f007:**
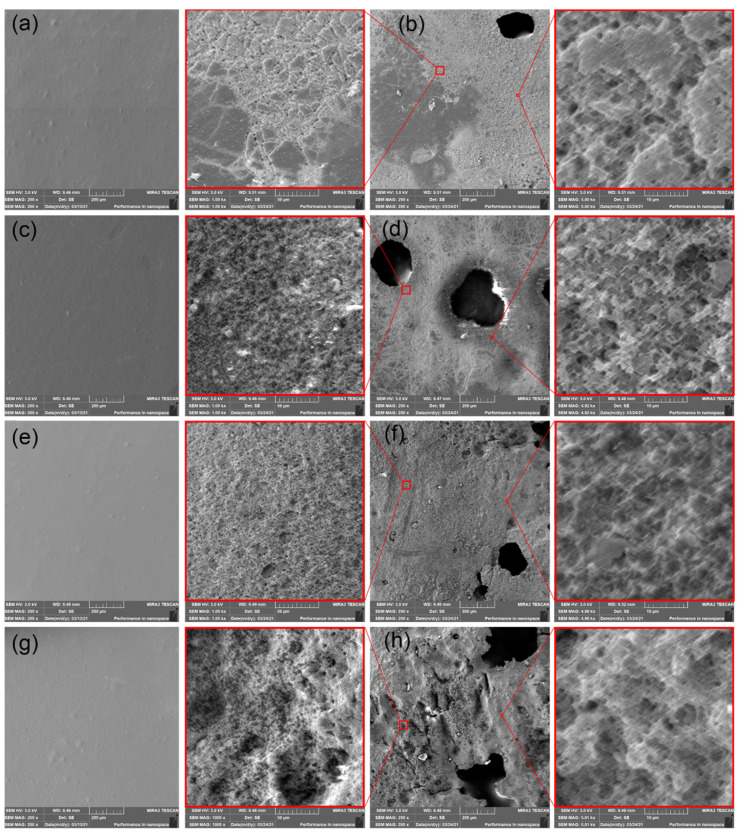
SEM images of (**a**) neat PHBV film at initial state, (**b**) neat PHBV film after 30 days of degradation in thermophilic composting, (**c**) PHBV/ATBC at initial state, (**d**) PHBV/ATBC film after 30 days of degradation in thermophilic composting, (**e**) PHBV/ATBC/L-CNC at initial state, (**f**) PHBV/ATBC/L-CNC film after 30 days of degradation in thermophilic composting, (**g**) PHBV/ATBC/CaCO_3_ at initial state, (**h**) PHBV/ATBC/CaCO_3_ film after 30 days of degradation in thermophilic composting.

**Table 1 polymers-14-00838-t001:** Sample compositions.

Sample Designation	Composition (wt. %)			
	PHBV	ATBC	L-CNC	CaCO_3_
PHBV	100	-	-	-
PHBV/ATBC	90	10	-	-
PHBV/ATBC/L-CNC	89	10	1	-
PHBV/ATBC/CaCO_3_	80	10	-	10

**Table 2 polymers-14-00838-t002:** The individual proportions of total organic carbon of PHBV biocomposite components.

Sample Designation	Proportions (%)
PHBV	55.8
ATBC	59.7
L-CNC	44.4
CaCO_3_	-

**Table 3 polymers-14-00838-t003:** Comparison of biodegradation level of PHBV and PLA bio-composites [48].

Sample Designation	PLA	PLA/ATBC	PLA/ATBC/ L-CNC	PLA/ATBC/ CaCO_3_	PBHV	PHBV/ ATBC	PHBV/ATBC/L-CNC	PHBV/ATBC/CaCO_3_
30 days	6%	16%	15%	58%	32%	35%	33%	56%
45 days	10%	31%	27%	83%	54%	73%	62%	88%
60 days	17%	69%	60%	94%	72%	88%	85%	98%
75 days	42%	97%	90%	100%	87%	92%	93%	99%

**Table 4 polymers-14-00838-t004:** Thermal analysis (DSC) data of PHBV bio-composites at initial state and after 30 days degradation.

Sample Designation	Exposition Time (Months)	T_cc_ (°C)	∆H_cc_ (J/g)	T_m1_ (°C)	T_m2_ (°C)	∆H_m_ (J/g)	X_C_ (%)
PHBV	Initial state	121.8	84.2	169.6	173.3	82.2	56
	Vermicompost	122.2	81.7	167.3	171.5	77.7	53
	Freshwater biotope	122.3	84.0	168.9	174.2	83.6	57
	Thermophilic composting	121.4	82.9	167.0	173.1	76.2	52
PHBV/ATBC	Initial state	119.7	81.2	166.4	170.1	80.5	61
	Vermicompost	119.6	80.4	169.2		79.2	60
	Freshwater biotope	118.5	73.8	162.7	169.9	72.6	55
	Thermophilic composting	119.6	78.3	165.3	170.5	73.9	56
PHBV/ATBC/L-CNC	Initial state	119.8	74.3	150.8	170.6	70.1	54
	Vermicompost	118.5	76.2	172.4		77.0	59
	Freshwater biotope	120.3	74.7	160.9	171.0	71.2	55
	Thermophilic composting	119.0	75.1	161.3	170.2	73.1	56
PHBV/ATBC/CaCO_3_	Initial state	119.1	65.5	160.6	169.5	62.2	53
	Vermicompost	118.8	69.8	170.8		68.0	58
	Freshwater biotope	119.9	64.5	161.2	170.3	59.8	51
	Thermophilic composting	118.3	67.6	162.2	169.9	63.3	54

**Table 5 polymers-14-00838-t005:** Thermogravimetric analysis (TGA) data of PHBV bio-composites at initial state, and after 30 days of degradation.

Sample Designation	Exposition Time (Months)							
	Initial State		Vermicomposting		Freshwater Biotope		Thermophilic Composting	
	T_3_ (%)	T_50_ (%)	T_3_ (%)	T_50_ (%)	T_3_ (%)	T_50_ (%)	T_3_ (%)	T_50_ (%)
PHBV	270.9	290.0	260.1	276.8	271.8	289.4	243.8	265.5
PHBV/ATBC	253.2	286.3	249.8	281.6	251.4	271.0	237.2	261.7
PHBV/ATBC/ L-CNC	254.6	287.1	251.1	284.8	257.7	284.9	244.3	275.8
PHBV/ATBC/ CaCO_3_	237.3	267.6	226.3	258.5	229.9	245.9	217.3	267.9

## Data Availability

The data presented in this study are available on request from the corresponding author.

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
