# Peer review of "The Influence of Additives and Environment on Biodegradation of PHBV Biocomposites"

_polymers, 2022, doi:10.3390/polym14040838_

Round 1
Reviewer 1 Report
The manuscript shows a comprehensive study of the degradation of PHBV-additive composites. The flow of the work is correct and the techniques used proved authors statements, the study is comprehensive and the techniques used are very well chosen. Just minor changes are required before publication and a couple of curiosities are asked.
Introduction: very well developed and complete. Maybe some information regarding PHBV scaffolds would be interesting to be added.
In table 1, why is the composition of PHBV 100% PLA, I believe this is a mistake… similarly wit table description of table 2.
Have you considered further studies of the enzymatic degradation (for example kinetic studies?)
Suggestion: remove references from the conclusion and add it on the discussion, just summarise the main outcomes in this section.
I would like to point out the interest in the comparing the present data with previous findings, as PLA present a lot of interest in this field. In addition, I would like to ask the authors to add a sentence in the conclusion that point out the interest of these results in potential applications of PHVB.
Author Response
Dear Sir,
Thank you very much for your revision and comments. I corrected the manuscript according to your advice. Answers to your comments are reported in the attachment.
with best regards
Pavel Brdlík

Reviewer 2 Report
The paper is on a topic of high interest, comparing different methods of biodegradation is a good approach in particular addressing relatively new methods such as vermi-composting, in this term the paper is innovative. There are several papers on PHBV with ATBC and CaCO3 also addressing thermal properties and biodegradation, but these are not referred in this paper while would be interesting to have a comparison discussion of the results. A moderate review of English is requested for some not clear sentences.
For the statement of CaCO3 coated with fatty acids inducing degradation on biopolyesters, more details, and eventually experimental evidences should be provided, and an explanation on why then using coated CaCO3 and not using raw CaCO3.
Methods: it is not clear how many replicates were done for the text and how much was the standard deviation among the replicates.
Figure 1, differentiate the line of the curves so that they may be readable even in white/black printing
TGA, the lower decomposition temperature in samples with OHBV and ATBC, may be due to ATBC degradation, this can be assessed doing a TGA of ATBC alone, or checking the data in literature form similar blends.
Avoid use of references in conclusions, this must be done in discussion of results, conclusion must be concise, while at present they are a summary of the results, you must extract only the “conclusion” of your results.
Supplementary material: “The not applicable” verb is missing…
Author Response

(The authors gave the same response as above.)
